# Meta-control of social learning strategies

**Anil Yaman**[1,5,6¤]*, **Nicolas Bredeche**[2], **Onur Çaylak**[3], **Joel Z. Leibo**[4], **Sang Wan Lee**[5,6,7,8,9]*

**1** Computer Science Department, Vrije Universiteit Amsterdam, Amsterdam, The Netherlands, **2** Institut des Systèmes Intelligents et de Robotique, Sorbonne Université, CNRS, Paris, France, **3** Department of Mathematics and Computer Science, Eindhoven University of Technology, Eindhoven, The Netherlands, **4** DeepMind, London, United Kingdom, **5** Department of Bio and Brain Engineering, Korea Advanced Institute of Science and Technology, Daejeon, Republic of Korea, **6** Center for Neuroscience-inspired AI, Korea Advanced Institute of Science and Technology, Daejeon, Republic of Korea, **7** Program of Brain and Cognitive Engineering, Korea Advanced Institute of Science and Technology, Daejeon, Republic of Korea, **8** KI for Health Science and Technology, Korea Advanced Institute of Science and Technology, Daejeon, Republic of Korea, **9** KI for Artificial Intelligence, Korea Advanced Institute of Science and Technology, Daejeon, Republic of Korea

¤ Current address: Computer Science Department, Vrije Universiteit Amsterdam, Amsterdam, The Netherlands
* a.yaman@vu.nl (AY); sangwan@kaist.ac.kr (SWL)

**Data Availability Statement:** Code is available at: https://github.com/anilyaman/Metacontrol-of-social-learning-strategies.

**Funding:** This work was supported by Institute for Information & Communications Technology Planning & Evaluation (IITP) grant funded by the

## Abstract

Social learning, copying other's behavior without actual experience, offers a cost-effective means of knowledge acquisition. However, it raises the fundamental question of which individuals have reliable information: successful individuals versus the majority. The former and the latter are known respectively as success-based and conformist social learning strategies. We show here that while the success-based strategy fully exploits the benign environment of low uncertainly, it fails in uncertain environments. On the other hand, the conformist strategy can effectively mitigate this adverse effect. Based on these findings, we hypothesized that meta-control of individual and social learning strategies provides effective and sample-efficient learning in volatile and uncertain environments. Simulations on a set of environments with various levels of volatility and uncertainty confirmed our hypothesis. The results imply that meta-control of social learning affords agents the leverage to resolve environmental uncertainty with minimal exploration cost, by exploiting others' learning as an external knowledge base.

## Author summary

Which individuals have reliable information: successful individuals or the majority? Seeking a suitable compromise between individual and social learning is crucial for optimum learning in a population. Motivated by the recent findings in neuroscience showing that the brain can arbitrate between different learning strategies, termed meta-control, we propose a meta-control approach in social learning context. First, we show that environmental uncertainty is a crucial predictor of the performance of the individual and two social learning strategies: success-based and conformist. Our meta social learning model uses environmental uncertainty to find a compromise between these two strategies. In

Korean government (MSIT) (No. 2019-0-01371, Development of brain-inspired AI with human-like intelligence), the National Research Foundation of Korea grant funded by the Korean government (MSIT) (No. NRF-2019M3E5D2A01066267), the NRF grant funded by the Korean government (MSIT) (No. 2021 M3E5D2A0102249311), IITP grant funded by the Korean government (No. 2017-0-00451), and Samsung Research Funding Center of Samsung Electronics under Project Number SRFC-TC1603-52. The main PI of these grants (corresponding author: SWL), and the research of AY has been supported by these grants whole time, and NB's participation is funded by the Agence Nationale pour la Recherche under Grant No ANR-18-CE33-0006. The funders had no role in study design, data collection and analysis, decision to publish, or preparation of the manuscript.

**Competing interests:** The authors have declared that no competing interests exist.

simulations on a set of environments with various levels of volatility and uncertainty, we demonstrate that our model outperforms other meta-social learning approaches. In the subsequent evolutionary analysis, we show that our model dominated others in survival rate. Our work provides a new account of the trade-offs between individual, success-based, and conformist learning strategies in multi-agent settings. Critically, our work unveils an optimal social learning strategy to resolve environmental uncertainty with minimal exploration cost.

## 1 Introduction

Learning is one of the basic requirements for animal survival. The ultimate goal of learning is to acquire reliable knowledge from a limited amount of interactions with the environment. However, the environment is often uncertain and volatile, making it difficult to learn.

Decades of studies found that animals have multiple learning strategies. For example, an animal can learn associations between environmental cues and outcomes (Pavlovian learning) or learn action-outcome associations (model-free reinforcement learning). It is a simple strategy but less adaptive to environmental changes because the action-outcome associations are gradually updated based on experience. A more sophisticated strategy is to learn the internal model of the environmental structure and to use this information to quickly perform actions in a more predictive manner (model-based reinforcement learning). While this strategy can accelerate the adaptation, encoding the internal model of the environment requires additional memory and computations [1, 2].

Recent studies in neuroscience suggest that the brain can find a compromise between these learning strategies via a process called meta-control [3–6]. Meta-control is based on the premise that learning strategies have different levels of sensitivity to environment variability and this can be measured by perceptual uncertainty concerning the association of actions and rewards [3]. Thus, perceptual uncertainty can be used to arbitrate between learning strategies. For example, in a stable environment, the brain prefers to use a sample-efficient model-based strategy, followed by a gradual transition to a computationally-efficient model-free learning strategy [7]. On the other hand, in environments with high perceived uncertainty, model-free learning are preferred over model-based learning because they are less susceptible to environmental uncertainty [8]. Accumulating evidence suggests that a part of the prefrontal cortex implements meta-control of various learning strategies, which provides a cost-effective solution to environmental uncertainty [1, 9, 10]. Ultimately, computational models of the brain's meta-control principle should find a way to efficiently avoid complications arising from environmental variability. [11].

Taking full advantage of the brain's meta-control capability of learning strategies, this paper proposes a meta-control approach to social learning, which we term *meta-social learning*. The proposed method aims to resolve environmental uncertainty by arbitrating between individual learning and two different social learning strategies, each of which exhibits a different uncertainty-sensitive performance-cost profile.

Both the individual and social learning strategies play a crucial role in learning as a population. Innovations are usually made by individual learning (IL) and spread throughout the population via social learning (SL) [12–14]. However, these strategies involve advantages and drawbacks, suggesting the need to trade social learning strategies off with individual learning [15, 16].

Individual learning can explore and discover useful innovations, however, it can be costly due to exploration, risk of injury, mortality, etc. [17]. It is only worthwhile to bear these costs if learning is required to adapt to the environmental changes. In static environments on the other hand, it would be unnecessary to pay these costs. To resolve this dilemma, one can suggest adapting the exploration rate so that it depends on the environment's variability (e.g. [18]). However, in this case, individual learners would still need to explore the action space by themselves to find the optimum behavior. This is inefficient if the optimum behavior was already discovered by other individuals in the population, and it can be readily copied. In that case it would be beneficial to make use of the knowledge explored by others in order to avoid paying the cost of exploring oneself.

In group-living animals in nature, social learning has evolved to take advantage of the exploration performed by others via copying their behavior (i.e. "free-riding" the exploration performed by the others without incurring the cost [19]), thereby reducing the cost of learning. Therefore, it does not involve these costs related to individual learning [20–23]. On the other hand, if there are too many individuals that take advantage of social learning then it may not be possible to explore other behaviors to find the optimum behavior especially in changing environments. In addition, social information can be less accurate since it depends on the observation of previously performed behaviors (i.e. may be outdated in case of environment change), and it requires identifying the individuals with reliable knowledge to copy. The term social learning strategy (SLS) refers to any of a variety of methods by which individuals can choose others to copy [20, 21, 24–26].

The efficiency of individual and social learning strategies in stable and dynamic environments has been demonstrated through theoretical and empirical studies [17, 27–30]. For instance in a computer tournament, Rendell *et al.* [31] noted the success of strategies that rely heavily on social learning over individual learning. They tested competing strategies on a fundamental decision-making problem known as the multi-armed bandit problem [32] (or $k$–armed bandit). Furthermore, they modeled a changing adaptive environment by adjusting the reward association of the arms during the task (it was a "restless" or "non-stationary" multi-armed bandit problem) [31, 33–35]. Despite these efforts, a fundamental issue in social learning still remains unaddressed: whether to use success-based social learning and copy the behavior of the successful individuals or simply follow the conformist strategy and copy the behavior that is the majority in the population. This poses a fundamental challenge for the individual, forcing them to confront a trade-off between performance and exploration cost [11].

To fully examine this issue, we performed an evolutionary analysis on individual learning and two social learning strategies, success-based and conformist [17, 21, 27, 36, 37] (see Fig 1). Despite several investigations into these strategies, the effect of environment uncertainty on their performance has remained largely unaddressed, making it hard to connect to the meta-control idea in neuroscience. To fill this gap, we designed non-stationary multi-armed bandit tasks with variable amounts of uncertainty. We modeled the reward distributions of the arms based on the Gaussian distributions, and introduced uncertainty into the tasks by the overlap between these distributions. In this settings, the uncertainty increases the difficulty of finding out the underlying distributions that these rewards are received from.

Individual learning was modeled as a value-based approach known as the $\epsilon$-greedy algorithm where the agents aim to figure out the arm that provides the highest average reward by trial-and-error [32] (see Section 4.2). In the case of social learning, success-based and conformist social learning strategies were modeled based on copying the actions of the most successful agent and majority of the agents respectively (see Section 4.3). This assumes that the success-based strategy has access to the knowledge of the action of the most successful

**Fig 1. The learning process of populations of individual and social learners on a binary decision-making task.** (a) A population of individuals perform a binary decision-making task based on individual and social learning strategies and collect their rewards ($r(a_j, t)$) based on their actions. The individual learners, modeled by a the $\epsilon$-greedy algorithm [32] (see Section 4.2), can perform their actions based on their decision models that can be improved by experience. The social learners use success-based or conformist strategies to copy the actions of successful individuals or the majority respectively (see Section 4.3). (b) Binary decision-making task (2-armed bandit) is iteratively performed for a certain period of time with specified reward distributions that are unknown to the individuals. At some point, an environment change occurs by changing the reward distributions (a.k.a reward reversal). In earlier stages of the process (initial and after environment change, shown in green dash lines), populations with success-based social learning strategy achieves higher average population reward faster relative to the conformist strategy. In later stages of the process (shown in purple dashed lines), populations with success-based social learning strategy achieves higher average population reward in environments with low uncertainty, whereas, populations with conformist social learners achieves higher average population reward in environments with high uncertainty.

individual (in terms of the rewards received), and the conformist strategy assumes the knowledge of the action that is performed by the majority of the individuals in the population. This is in line with the hypothesis that suggests that humans, in particular, possess domain specific cognitive capabilities for social learning to allow them to assess the knowledge and performance of others [5, 16, 21, 38]. Our setup can be used as an abstracted model of fundamental decision making processes in nature, such as foraging, predator avoidance, symbiosis, and mutualism [26, 39, 40], as well as human decision making processes, such as human-robot interactions, investment decisions in stock markets, consumer decision making, dynamics of social networks, etc.

Our simulations confirmed the view that while the success-based strategy is vulnerable to environmental uncertainty, the conformist strategy serves as an alternative that can effectively resolve the adverse effect of uncertainty. These results show that neither individual learning, success-based, nor conformist social learning strategies are on their own sufficient to achieve a optimal policy for lifetime learning. This view motivates us to hypothesize that there exists an ideal combination of these strategies to cope with the environment volatility and uncertainty. In doing so, we propose a meta-social learning strategy that uses estimated uncertainty to arbitrate between individual and social learning strategies.

To test our hypothesis, this model was pitted against a large set of algorithms implementing various strategies based on reinforcement learning [32], genetic algorithms [41] and neuroevolution [42, 43] in environments with various levels of volatility and uncertainty. The results demonstrate that the proposed model serves to achieve near-optimal lifetime learning in the sense that it resolves the performance-exploration cost dilemma. We also show that the version

of the models implementing our hypothesis tends to have a higher ratios in the populations and survive longer relative to the others throughout our evolutionary analysis.

## 2 Results

### 2.1 Uncertainty-invariance in social learning

To examine whether and to what extent environmental uncertainty influences performance of individual and social learning strategies, we compared adaptation performance of various types of learning strategies in different levels of environmental uncertainty. We considered independent populations consisting of individual learners using the success-based social learning strategy, and individual learners with the conformist strategy. To model the evolutionary dynamics of these populations, we used two distinct approaches: (1) a mathematical model based on the replicator-mutator equation [44, 45], and (2) an agent-based evolutionary algorithm [41]. These approaches allow us to tract the change in the ratios of the individual and social learners within a certain environment. Fig 2 provides a summary of these results. For detailed analysis of the population dynamics, and the stability analysis and basin of attraction, see Section S1.1 in S1 Text. In addition, in Section S1.2 in S1 Text, we compare the performance of success-based and conformist strategies with two social learning cases where individuals copy a perfect and random model. These cases do not require the knowledge of the successful individual and the decision of the majority.

In case of the mathematical model, the change in the frequencies of individual/social learners selecting a given arm are modeled by a system of coupled first order differential equations. This approach has largely been used in evolutionary game theory [46, 47]. The fitness of the types of individuals was defined based on the rewards received. Note that the individual learners carry a constant computational cost of learning. To the contrary, the social learners avoid this issue by simply copying the others' choice, although they are deemed to endure a latency issue arising from the necessity of observing the past choices of the others.

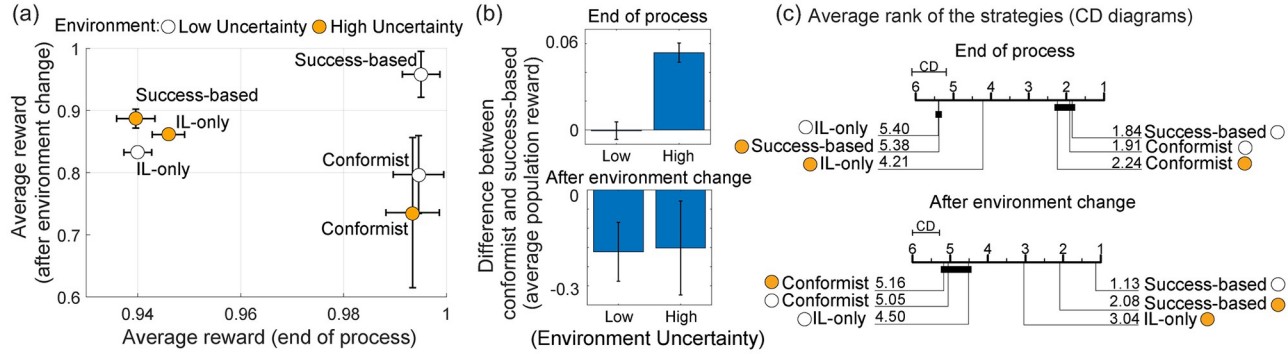

**Fig 2. Performance of individual and success-based and conformist learning strategies.** The success-based strategy shows the best performance in environments with low uncertainty, however, it suffers when there is uncertainty in the environment. On the other hand, the conformist strategy achieves similar performance independent of the environment uncertainty. (a) The average rewards achieved by the strategies after an environment change vs. at the end of the processes in environments with high and low uncertainty. (b) The difference in the performance of conformist vs success-based strategies. After an environment change, the performance achieved by the success-based strategy is higher in low and high uncertainty environments, whereas, at the end of the processes, they achieve similar results only in low uncertainty environment. On the other hand, in high uncertainty environment the performance of the conformist strategy is higher. Finally (c) provides the Critical Difference diagrams (CD) shows the average ranks of the the strategies (given by decimal numerals in each line that represent a strategy) based on their performance after an environment change and at the end of the processes. Lower ranks demonstrate better performance in terms of average population reward achieved by the strategies. The strategies are linked (via bold thick lines) if their average ranks are not statistically significant based on the post-hoc Nemenyi test at $\alpha = 0.05$ [48, 49].

In case of the agent-based evolutionary algorithm, we simulate the evolutionary process involving a population of individual and social learners, each of which were modelled separately. The individual learners have the capacity to improve their behavioral policy over time based on their experience. They were implemented using the $\epsilon$-greedy algorithm [32] in which an exploration parameter $\epsilon$ is the probability per timestep of taking a random (uniformly distributed) action instead of taking the current greedy action with the highest average reward. Note that this exploration carries an extra cost when the individual is already making an optimal choice (see Section 4). The social learners, on the other hand, perform their actions by copying others with a certain level of latency. This process is repeated in each generation where all the individuals made their choices and receive their rewards. At the end of each generation cycle, individuals were sampled with replacement for the next generation proportional to their fitness.

An example illustration of a binary decision-making task given in Fig 1a and 1b. The reward distributions are parameterized by Gaussian distributions with a mean ($\mu$) and standard deviation ($\sigma$). Since the performance of success-based social learning is contingent on correctly identifying agents making optimal choices, we hypothesized that the high uncertainty in the environment, introduced by a larger overlap between the reward distributions as illustrated in Fig 1b, would make it hard to identify successful individuals, leading to the degradation of its learning outcomes. To test this, we designed novel tasks with different levels of uncertainty, controlled by the degree of overlap between reward distributions of different arms. The fitness values of the individuals are defined as the amount of rewards received following their choice. To assess the individuals' adaptation ability to environmental changes, at the midpoint of each simulation we reversed the association between arms and their reward distributions.

Fig 2 shows the performance comparison of the populations consisting of only individual learners, success-based and individual learners, and conformist and individual learners after the environment change (the average of the period between $t = [200, 250]$) and at the end of the process (average of the period between $t = [350, 400]$). To formally quantify the effect of uncertainty on performance, we also used critical difference (CD) diagrams. The critical difference (CD) diagrams allow comparison of results from multiple strategies. They show the average ranks of the algorithms from the best to the worst. The algorithms that do not have significant rank difference are linked (post-hoc Nemenyi test [48] at $\alpha = 0.05$ [49]). The population dynamics throughout the evolutionary processes are provided in Section S1.1 in S1 Text. When the uncertainty in the environment is low, the success-based social learning strategy shows the best performance in terms of adaptation after an environment change ($p < 0.01$; Wilcoxon rank-sum test [50]), and at the end of the process, the success-based and conformist strategies show similar performance, and they are superior to individual learning only. However, when the uncertainty in the environment is increased, populations with conformist social learners achieve higher average population reward ($p < 0.01$).

The evolutionary parameters, in this case mutation rate, is expected to have an effect in the evolutionary dynamics [51]. We performed sensitivity analysis for various assignments of mutation rate and show the results in Section S1.3 in S1 Text. We observe the following effect that is in line with our conclusions: the higher the mutation rate, the higher the rate in which randomly mutated individuals (as individual or social learners) are introduced into the populations. Since randomly mutated individuals would not perform optimally, higher mutation rates cause a reduction in the performance of the populations with social learners.

We performed an additional analysis on a case where the reward functions are binary. This scenario would not involve uncertainty in the same way studied in this work (i.e. given by the overlap between the Gaussian reward distributions). However, the reward distributions of the

arms can be modeled to provide binary rewards (i.e. without the loss of generality, the arms can either provide 1 or 0, or two other arbitrary reward values) with certain probabilities $\mu_1$ and $\mu_2$, and in this case, smaller differences between these probabilities cause higher uncertainty in identifying successful individuals in the population to copy. We performed simulations for this case and presented in Section S1.4 in S1 Text. Our simulations showed that decrease in the difference between $\mu_1$ and $\mu_2|$ causes the performance of the success-based strategy to decrease. On the other hand, conformist strategy achieves a higher average population reward relative to success-based strategy, and shows robust performance independent of $|\mu_1 - \mu_2|$. We note that the results of this additional analysis are aligned fully with our results. Therefore, it can provide additional evidence and further support for our conclusions regarding the effect of the environment uncertainty to success-based strategy.

## 2.2 Uncertainty as a predictor of social learning performance

To further investigate the relationship between environmental uncertainty and social learning strategies, we measured the performance difference between the success-based and conformist strategy as a function of the amount of uncertainty in reward distributions. We refer to this uncertainty as the optimum distribution prediction uncertainty (ODPU) because it undermines the ability of the success-based strategy to correctly identify individuals making optimal choices. We computed the ODPU directly by the probability of receiving better rewards from the sub-optimal reward distributions. For example, if the ODPU is high, it is more likely to mistake an individual making sub-optimal choice as a successful individual and copy its action.

Considering a population consists of $M$ and $N$ individuals making choices to collect rewards from the environment associated with certain reward distributions. In this case, the ODPU depends not only on the sufficient statistics of the reward distributions but also on the size of the subgroup of individuals making optimal and sub-optimal choices, denoted by $M$ and $N$, respectively. In Fig 3, we illustrate the performance difference between the success-based and conformist strategies as a function of the ODPU on two Gaussian reward distributions. The generalized version of the ODPU that can be applied to more than two reward distributions is provided in Section 4.4.

In Fig 3c and 3d, we show the ODPU when $M = 5$, $N = 95$ and $M = 50$, $N = 50$ depending on $\sigma_1$, $\sigma_2$ respectively. Overall, the smaller the number of individuals making optimal choices, the larger the ODPU. In addition, an increase in $\sigma_1$ and a decrease in $\sigma_2$ causes an increase in the ODPU. Fig 3e shows the strong correlation between the ODPU and the difference between performance of the conformist and of the success-based strategy (with Pearson's correlation coefficient $r = 0.9791$). Note that after a certain ODPU value (i.e. approximately around 0.1–0.2 in Fig 3e), their performance difference becomes highly significant. The maximum possible performance difference depends on the difference in their $\mu$.

## 2.3 Meta-social learning hypothesis

In this section, we propose meta-social learning as a way for agents to arbitrate between individual and social learning strategies during their lifetime. We explored this hypothesis using several approaches. First, we used context encoding approach to determine the "context" of the environment by estimating three environmental variables, namely, environment change ($EC(t)$), conformity ($C(t)$) and uncertainty ($U(t)$) that play crucial role in the performance of the individual and social learning strategies. Then, based on our analysis, we defined meta-social learners that can arbitrate between these strategies depending on the context of the environment. Furthermore, as alternative approaches, we used evolutionary algorithms and

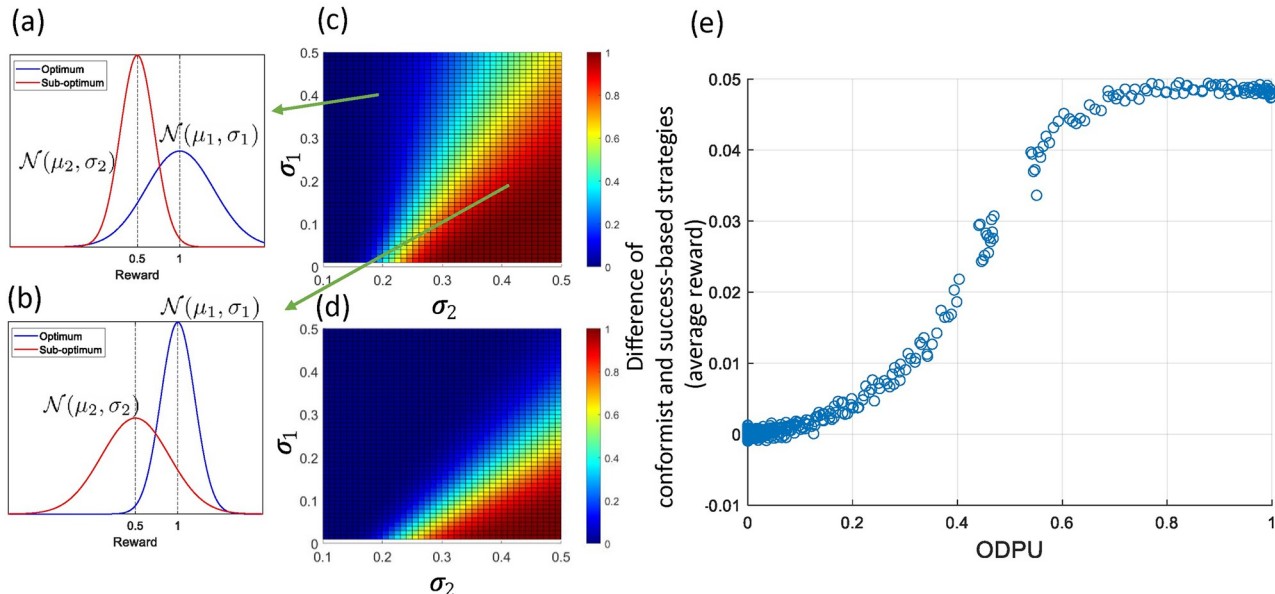

**Fig 3. The higher the uncertainty between two distributions (measured by the ODPU) the higher the performance difference between conformist and success-based strategies in terms of average population reward at the end of simulation processes.** (a) and (b) illustrate two cases where $\sigma_1 = 0.4$, $\sigma_2 = 0.2$ and $\sigma_1 = 0.2$, $\sigma_2 = 0.4$ respectively. (c) and (d) show the ODPU, formalized as the probability of sampling the highest reward value from the sub-optimum distribution, depending on $\sigma_1$ and $\sigma_2$ and the ratios of samples drawn independently from the optimum and sub-optimum reward distributions. In (c) the ratios of samples drawn from the optimum and sub-optimum distributions are 0.05 and 0.95, and in (d) the ratios are 0.5 and 0.5 respectively. (e) shows the relation between the ODPU and the difference in average reward at the end of the process between the populations with conformist and success-based strategies.

reinforcement learning to optimize the meta-social learners. Finally, we defined several baseline strategies that perform individual and social learning strategies randomly with a predefined probabilities.

**2.3.1 Context encoding.** We utilize the "social information" in estimating environment change, conformity and uncertainty. the social information is assumed be available for all individuals in the population and it consists of the action distribution of the population $h(a_j, t) \in \boldsymbol{H}$ and the rewards received by the individuals $r_i(a_j, t) \in \boldsymbol{R}$ where $h(a_j, t)$ denotes the frequency of action $a_j$ in the population and $r_i(a_j, t)$ denotes the reward of individual $i$ by performing action $a_j$ at time $t$. From the reward distribution, it is possible to estimate average rewards $\mu'_j(t)$ and standard deviations $\sigma'_j(t)$ of actions $a_j$. Note that success-based social learning requires finding the action with the highest reward and conformist strategy requires finding the action with the highest frequency, therefore the social information is the same as the information required to perform the success-based and conformist strategies.

**Environment change.** It is defined as the difference between the current and previous values of average rewards of the estimated optimum action:

$$EC(t) = \begin{cases} 1, & \text{if } |\mu'_*(t) - \mu'_*(t - \delta)| > th_{ec}, \\ 0, & \text{otherwise.} \end{cases}$$

where subscript $*$ denotes the estimated optimum action (that is the action with the highest average reward), $\delta$ is a parameter for comparing previous values of the average rewards, and $th_{EC}$ is threshold for triggering the environment change detection. Threshold $th_{EC}$ can depend on the task. In our experiments, we assign 0.15 for this threshold.

**Conformity.** It is based on the estimation whether the majority of the individuals are performing the behavior with the highest average reward. Thus, it is defined as:

$$C(t) = \begin{cases} 1 \ (\text{conformity}), & \text{if } \arg\max_j \ \mu'_j(t) = \arg\max_j \ h(a_j, t), \\ 0 \ (\text{non} - \text{conformity}), & \text{if } EC(t) = 1, \\ 0 \ (\text{non} - \text{conformity}), & \text{otherwise}. \end{cases}$$

Furthermore, if an environment change is detected, conformity is reset to 0.

**Uncertainty.** Estimated based on the ODPU (the probability of sampling higher reward values from sub-optimum distributions, see Sections 2.2 and 2.4). We use average rewards $\mu'_j(t)$ and standard deviations $\sigma'_j(t)$ to compute the ODPU. Then, the uncertainty is detected as:

$$U(t) = \begin{cases} 1 \ (\text{high uncertainty}), & \text{if } ODPU > th_u, \\ 0 \ (\text{low uncertainty}), & \text{otherwise}. \end{cases}$$

where uncertainty threshold $th_u$ is set to 0.1 in our experiments based on our uncertainty analysis in Section 2.2.

**2.3.2 Meta-control of social learning strategies.** A generic representation of the strategy selection process of a meta-social learner is shown in Eq 1. A strategy $S \in \{$*individual learning, success-based and conformist*$\}$ is selected by meta-social learner $MSL()$ based on the context of the environment.

$$S := MSL(EC(t), C(t), U(t)) \tag{1}$$

In addition, we implement several other meta-social learning control mechanisms using various approaches and discuss under four types as follows (for implementation details of these algorithms, see Section 4.6):

**Observation-based control:** Here, we implement three versions. All of these versions start the process by using individual learning strategy. Similarly, they switch back to individual learning after an environment change. Otherwise, they use success-based or conformist social learning depending on the conformity and uncertainty.

- SL-EC-Conf (uses environment change and conformity) switches to the conformist social learning strategy if conformity ($C(t)$) is satisfied.

- SL-EC-Succ (uses environment change and uncertainty) switches to the success-based strategy in low uncertainty environment. Otherwise, they use individual learning.

- SL-EC-Conf-Unc (uses environment change, conformity and uncertainty) arbitrates between social learning strategies depending on conformity and uncertainty. If the conformity is observed in the population, then the individuals switch to the conformist strategy. Otherwise, if the environment is with low uncertainty, then they use the success-based strategy. If none of the above conditions is met, they perform individual learning.

**Evolutionary control:** The control policies for arbitrating between individual and social learning strategies were achieved by evolutionary algorithms. We used different environments (provided in Sections S1.6 and S1.7 in S1 Text) for training and testing. We performed evolutionary-based training for 10 independent runs and selected the best performing strategy. Then, we tested this strategy on the test environments that were not encountered during the

training processes and reported the results. The goal of this separation was to demonstrate the generalization capability of the train model. We implemented two versions.

- SL-GA (trained by the genetic algorithms) perform the task based on the rules optimized with the genetic algorithms [41]. These rules are optimized to select individual and social learning strategies depending on the binary states of $EC(t)$, $C(t)$ and $U(t)$ (see Section 4.6.2). It is possible to explore all possible rules (based on all possible states of $EC(t)$, $C(t)$ and $U(t)$) in this space to identify the "optimal" rule. We perform 10 independent evolutionary runs and select the best control mechanism based on their cumulative reward. The results of the training processes are provided in Section S1.6 in S1 Text. We note that the best performing evolutionary control mechanism found by SL-GA converged to SL-EC--Conf-Unc.

- SL-NE (artificial neural network trained by neuroevolution) utilizes an artificial neural network to control meta-social learning, whose parameters were optimized by an evolutionary algorithm (known as neuroevolution approach [43]). The fully connected feedforward network (FCN) with one hidden layer takes the average rewards of arms $(\mu'_1(t), \ldots, \mu'_k(t))$, standard deviations $(\sigma'_1(t), \ldots, \sigma'_k(t))$ and the frequencies of the individuals that select $k$ arms $(h(a_1, t), \ldots, h(a_k, t))$, and chooses a strategy as follows:

$$S := FCN(\mu'_1(t), \ldots, \mu'_k(t), \sigma'_1(t), \ldots, \sigma'_k(t), h(a_1, t), \ldots, h(a_k, t)) \qquad (2)$$

Note that unlike other versions of meta-social learning, this one does not require identifying the context of the environment such as environment change, uncertainty or conformity.

**Multi-armed bandit control:** We used SL-RL ($\epsilon$-greedy algorithm), SL-UCB (upper confidence bound algorithm) and SL-QL (Q-learning) algorithms to learn to choose between individual, success-based and conformist social learning strategies. SL-RL and SL-UCB does not use environment context, rather, they perform a strategy selection based on the estimated rewards of selecting these strategies. The estimated rewards of these strategies are updated based on the rewards received after their selection. The SL-QL uses a value learning, intended to maximize the expected amount of future rewards. We used the context of the environment (environment change, uncertainty and conformity) as the states.

**Other baseline strategies:** We further implemented a set of baseline strategies: SL-Rand, SL-Prop, SL-Conf, SL-Succ, and IL-Only that perform a random strategy with a predefined fixed probability (see Section 4.6 for details). These meta-social learning strategies do not make use if the context of the environment.

## 2.4 Uncertainty based meta-control resolves the trade-off between performance and exploration cost

To compare the performance of the meta-social learning algorithms on a task with uncertainty changes, first we defined stable and volatile environments with low and high uncertainty (see Section S1.8 in S1 Text). Here, stable and volatile environments refer to the number of changes, namely, two and five respectively, performed in the reward distributions during the evolutionary processes. To construct these environments, we arbitrarily generated a set of six reward distributions from the highest to the lowest levels of uncertainty. Then, we created a task consisting of multiple periods, each of which is associated with a reward distribution selected from this set (**Experiment1**). Moreover, we ran additional tests with two challenging tasks: random volatile environment where the number of environment changes and distributions were chosen randomly (**Experiment2**), and uncertain environment with a gradual

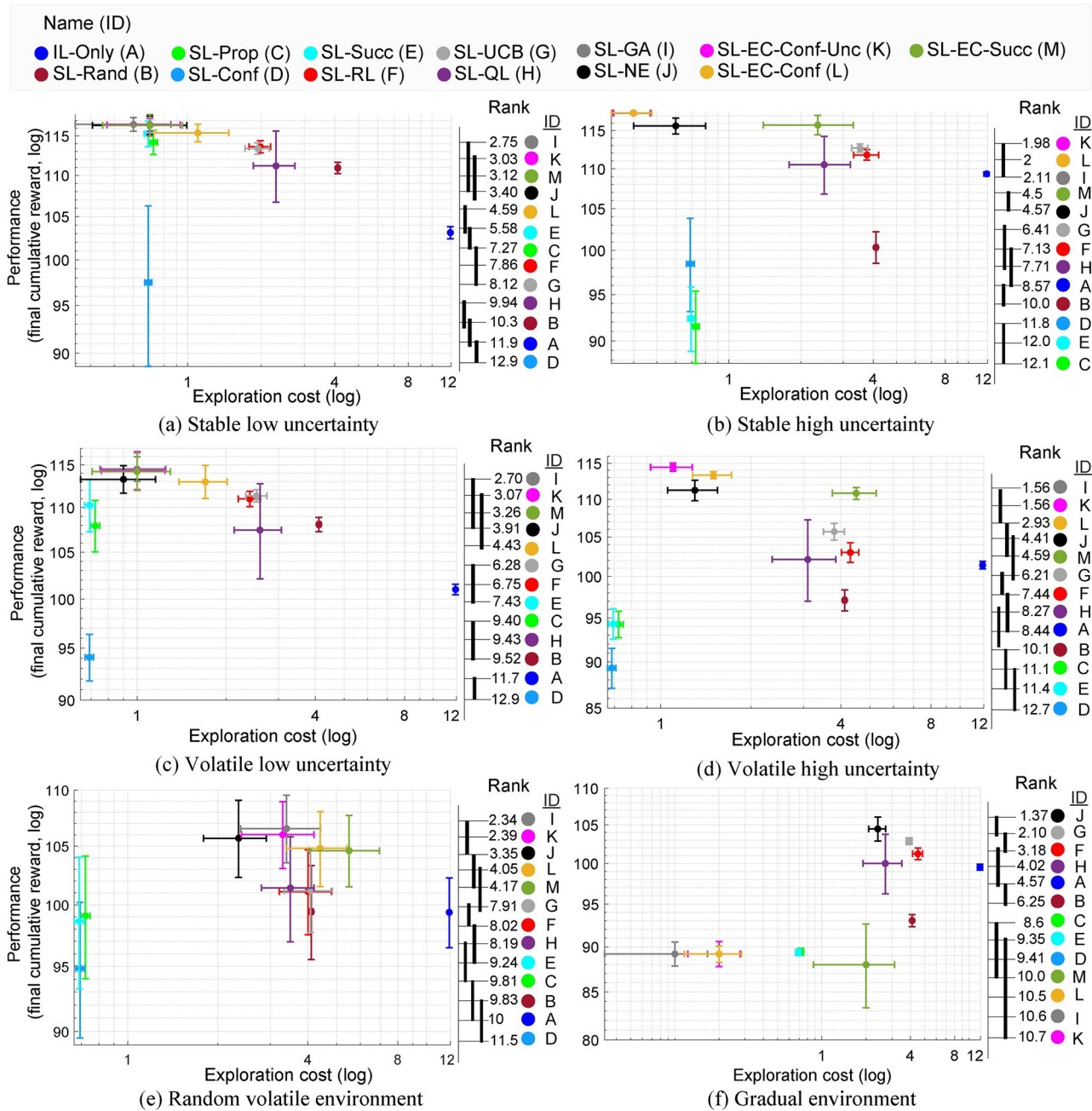

**Fig 4. SL-GA and SL-EC-Conf-Unc show the best performance vs. exploration cost on diverse set of environments. Overall, the meta-social learning strategies that utilize conformist strategy show better performance environments with high uncertainty.** On the right of each figure, the color coded labels indicate the meta-social learning strategies (A through M maps to the strategy with the same color shown on top), and are ranked based on their performance from the best to worst. The decimal numerals on the left of the color codes indicate their average ranks (the lower is the better). The strategies are linked with vertical bold lines if the differences in their average ranks are not statistically significant at $\alpha = 0.05$ based on the post-hoc Nemenyi test.

environment change (**Experiment3**). The performance-exploration cost trade-off and their ranks (based on CD diagrams) are shown in Fig 4. Exploration is provided only through individual learners and controlled by $\epsilon$, therefore, we measure the exploration cost by the number of individual learners used throughout a process multiplied by their $\epsilon$. The change of the

average reward and cumulative average reward during the learning processes on these environments are shown in Section S1.8 in S1 Text.

Overall, both the SL-GA and SL-EC-Conf-Unc achieved the highest performance with lowest exploration cost (the performance of the two models are not significantly different; Fig 4). This is due to the fact that the evolved controller in SL-GA is converged to the same controller used in the SL-EC-Conf-Unc. We note that the SL-NE provides one of the top five ranking results even though it uses low level population based features with artificial neural networks.

In general, the models utilizing the conformist strategy showed better performance in uncertain environments, compared to the ones with the success-based strategy. From the exploration cost point of view, we note that the IL-Only suffers from the highest cost with about three times more costly than the second most costly meta-social learner. In the case of the uncertain environment with gradual environment change (**Experiment3**; see Fig 4f), it is not surprising that the algorithms relying on the threshold-based environment change detection (SL-EC-Conf, SL-EC-Succ, SL-EC-Conf-Unc) did not perform well, which is ascribed to the failure in detecting environment change. To the contrary, SL-NE showed reliable performance robust against environment changes even though it was trained on the environments with different conditions. Note that this remarkable adaptation ability does not require an explicit environment change detection mechanism.

## 2.5 Uncertainty based meta-control dominates the evolution in volatile environments

What if there is a competition between different meta-social learning strategies in environments with various levels of volatility and uncertainty? Which ones would persist in the populations and become dominant relative to others? To assess that, we conducted an evolutionary analysis on meta-social learning. We further recorded how long they stay in the population (age) to assess their resilience. This analysis shows us successful strategies that are not being invaded by other strategies even in changing environmental conditions.

In the beginning of the evolutionary processes, we assigned each individual a specific type of meta-social learning strategy, randomly sampled from the complete set of the social learning strategies (i.e. IL-Only, SL-Rand, SL-Prop, . . ., SL-EC-Conf-Unc). The individuals then used their own meta-social learning strategy to perform the tasks. After each generation, meta-social learners are selected based on the probability proportional to their fitness values (rewards received in the previous generation). Furthermore, we introduce a mutation operator that re-samples the type of meta-social learning strategy of each individual at each generation based on a small probability controlled by mutation rate *mr*. At every generation, the age of all strategies is increased by one. It is possible to pass multiple copies of a strategy to the next generation during the selection process. In this case, multiple copies are treated as offspring where only the age of the first copy is preserved while the age of the others is set to zero. Similarly, after mutation the age of the strategy is set to zero.

Fig 5 shows the population dynamics of meta-social learning algorithms during the evolutionary processes. For statistical significance, we used a population consisting of 5000 individuals and perform the evolutionary processes for 112 independent runs. The statistical significance of the results measured by pairwise *p*-values and differences that are not statistically significant shown on the right side of each figure. Furthermore, we provide the standard deviations of the results of multiple runs highlighted on each figure or shown in form of error bars. The meta-social learning strategies that perform well relative to the others show increase in their ratios in the population, whereas, the ones that cannot perform well show decrease in

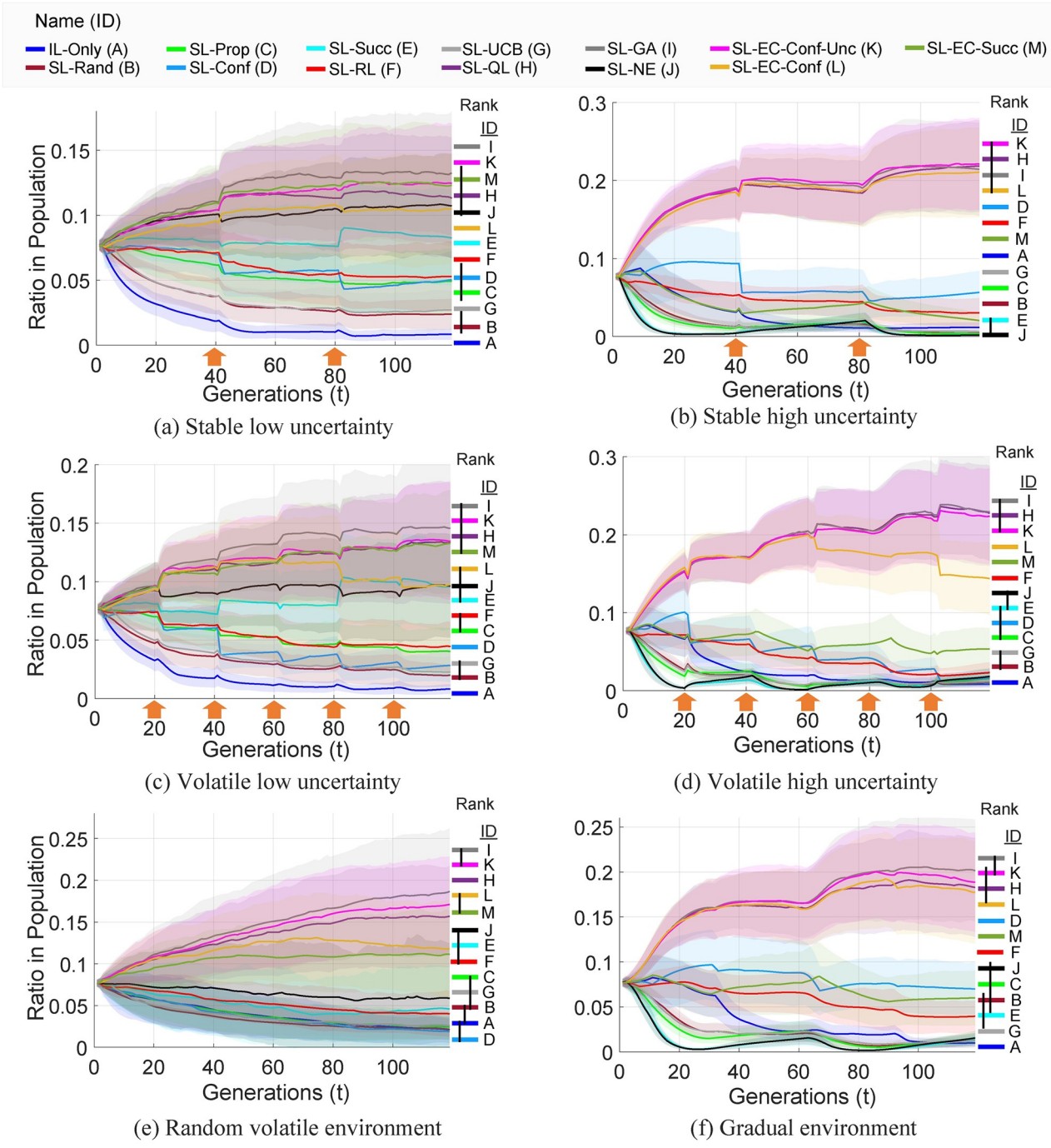

**Fig 5. Based on the ratios in the populations, SL-GA, SL-EC-Conf-Unc, SL-QL and SL-EC-Conf are the most dominating meta-social learning strategies in wide range of environments.** On the right of each figure, the ranks of the algorithms (higher to lower in terms of ratios) at the end of the processes are shown. The strategies were labeled and color coded to correspond to names provided on top. The differences that are not statistically significant (at $p > 0.05$, Wilcoxon rank-sum test) are linked by black vertical lines. The points of environment change are indicated with orange arrows. The highlighted areas show the standard deviation 112 runs.

their ratios, and eventually die out however due to the use of mutation (with rate of 0.005), strategies are not completely eliminated from the evolutionary processes.

Overall, in the environments with low uncertainty, seven meta-social learning strategies show an increase in their ratios at the end of the processes relative to their starting ratios,

whereas, in the environments with high uncertainty, only four of them show increase in their ratios. The most dominating four meta-social learners are: SL-EC-Conf-Unc, SL-GA, SL-EC-Conf and SL-QL. These meta-strategies, with the exception of SL-EC-Conf, make use of the environment uncertainty information. Remarkably, SL-EC-Conf is able to compete with the others using conformity bias without the need of using environment uncertainty.

We note that even though meta-social strategies such as SL-GA, SL-EC-Conf-Unc that failed to perform well in gradual environment (see Fig 4f) by themselves, they can show domination over other strategies in this experiment (see Fig 5f). This is due to the fact that the populations that consist of only these strategies cannot detect the environment change leading to the failure of exploring the other arm after an environment change. However, when they are used in combination with the other meta-social strategies (e.g. SL-Rand, SL-IL, SL-NE, . . .), the exploration performed by other meta-social strategies helps overcome this effect. Consequently, these strategies (that fail due to the environment detection mechanisms) can perform well and dominate the populations.

We performed an analysis on the age distributions of the meta-social learning strategies during the evolutionary processes (see Section S1.9 in S1 Text). Sudden changes in the age distribution due to the environment change can be observed. The dominant meta-social learners show higher life expectancy throughout the processes especially in the environments with high uncertainty.

We further hypothesize that the life expectancy of the dominant meta-social learning strategies should be consistent with key variables of the evolutionary process, such as the selection strength and mutation rate. To test this we performed further experiments, in which we ran the same simulations while varying the level of selection strengths and mutation rates (as *low*, *moderate* and *high*). The selection strength is given by $p_i = f_i^s / \sum_{j=1}^{m} f_j^s$, where $p_i$ is the selection probability of a meta-social learning strategy $i$, to pass to the next generation, $f_i$ is its fitness value, and $m$ is the total number of meta-social learners.

While the selection strength increases, the life expectancy of the dominant strategies increases, whereas, the life expectancy of the others decreases (see Section S1.10 in S1 Text). Furthermore, it can be observed that, while the mutation rate increases, the life expectancy of the dominant strategies decreases. This is due to the fact that, when the mutation rate is high, the probability of randomly mutating a dominant strategy increases, causing their life expectancy to decrease.

## 3 Discussion

While previous research examined individual and social learning strategies in the context of a changing environment [17, 27–31], this study tested a new hypothesis that measurements of environmental uncertainty (formalized by the UDPU) can be used as a means to implement a reliable and cost-efficient learning strategy, regardless of environmental changes. To test this hypothesis, we performed an analysis on individual learning and two social learning strategies, namely success-based and conformist, on volatile and uncertain environments. Our analysis showed that the performance of the success-based strategy, the most direct way to explore to find the optimal policy, is susceptible to uncertainty in the environment, whereas that of the conformist strategy, though it does not guarantee the optimal performance, is highly reliable. Motivated by these results, we proposed several meta-social learning algorithms. Overall, the proposed meta-social learning strategies showed significantly better performance, and in the evolutionary analysis, they dominated other meta-social learning approaches in terms of survival rate.

The proposed meta-social learning scheme is motivated by the recent theoretical idea in neuroscience, suggesting that the brain uses meta-learning strategies to find a compromise between different types of learning, such as Pavlovian, model-free, and model-based learning [3, 4, 10, 52]. Accumulating evidence suggests that meta-learning allows individuals to resolve environmental uncertainty efficiently [1, 9, 11, 15, 53]. Critically, this view is fully consistent with our finding that meta-social learning strategy mitigates the adverse effect of environmental uncertainty on performance. Our computational framework can thus be used to examine neural mechanisms of meta-social learning [5].

The success-based and conformist social learning strategies modeled in this work assume that the individuals have access to the knowledge of the action of the most successful individual, and the action performed by the majority respectively. One interesting approach is to model the learning mechanisms to learn based on the choices of other agents (e.g. observational learning [15, 53]). Since the prior works were concerned with exploring key variables, albeit in limited settings, to determine an individual to copy or infer underlying goals, our framework could be extended by incorporating these features.

Our neural network based meta-control strategy (SL-NE) can be viewed as a form of observational learning. In this case, our network takes the frequencies of actions and rewards received by agents in the population as inputs and decides which action to take. More elaborate version of this model can identify which individual to copy by looking at the actions and rewards of all individuals. However, testing these models should outweigh various computational issues. For example, the addition of such features essentially increases the size of the input space for the same task, and may not provide further understanding of the optimality of the social learning strategies depending on the environment uncertainty as studied in this work. We acknowledge that understanding neural mechanisms to identify individuals to copy their behavior or infer their goals based on population dynamics is a very interesting research direction. Future works could directly benefit from our meta-control social learning frameworks. For example, variants of the SL-NE can be used to explore new neural hypotheses in more general settings.

Our meta-social learning framework provides a means to examine complex population dynamics, thereby helping us better understand the fundamental nature of biological learning and decision making [15, 26, 39]. For example, our meta-social learning principle would provide theoretical insight into why animals or humans often copy others' behavior and why society needs to achieve conformity, especially in highly volatile situations. It would also be possible to examine how animal societies cope with environmental uncertainty and volatility. Moreover, the meta-social learning scheme can be extended to explain various types of ecological interactions, such as symbiosis, mimicry, mutualism and cooperation [54].

In artificial intelligence and robotics applications, nature inspired approaches have proven to be successful in modeling intelligent behavior [55–58]. Accordingly, social learning aims to benefit from the collective property of multi-agent systems to provide efficient learning and adaptation as a population. As illustrated in this work, exploiting the behaviors of other individuals can reduce exploration cost significantly. It can also improve learning efficiency in uncertain and volatile environments. This may prove to be important in real-world applications such as the internet of things and swarm robotics [59–63]. For instance, one recent example of distributed learning approach has been used in healthcare to detect illnesses [64]. Meta-social learning strategies can play a key role in these kinds of distributed learning applications to improve the efficiency of learning.

Rational choice theory in economics and game theory suggests that individuals choose their best action through a cost-benefit analysis which we usually conceptualize as involving explicit deduction (thinking through pros and cons) [65, 66]. Since our results suggest that an individual can make cost-effective decisions instead via social information, i.e., the decisions of others

and their outcomes, it may be useful to consider models based on such foundations as well. For instance, it would be possible to estimate the environment uncertainty and volatility simply by measuring individual choice variability. This inference based on social information improves sample efficiency significantly compared to individual learning.

In addition to the computational and theoretical implications of meta-social learning, another exciting research direction is to use meta-social learning to examine the fundamental nature of social networks [67]. For instance, complex dynamics of individual interactions can lead to the emergence of various "social learning networks". Investigation of fundamental computations underlying the emergence and evolution of social networks would allow us to understand and predict the future of animal societies. Furthermore, the computational framework of meta-social learning be used to test new hypotheses about multi-agent social learning [68, 69]. For instance, it is possible to test whether complex internal dynamics arising from meta-social learning promote the natural emergence of curriculum.

## 4 Methods

### 4.1 Multi-armed bandit problem

The multi-armed bandit problem is a classic problem in reinforcement learning that [31, 32, 34] where individuals are required to perform actions to choose one of $k$ alternative choices (also known as $k$-arms). Performed actions provide rewards based on their underlying distributions that is unknown to the individual. In our work, We model the reward distribution of each action as a Gaussian distribution $\{\mathcal{N}(\mu_1, \sigma_1), \dots, \mathcal{N}(\mu_k, \sigma_k)\}$. The goal of an individual is to perform actions to choose repetitively one of the choices and collect the rewards for a certain period of time such a way that can maximize the cumulative sum of the rewards received during the process.

### 4.2 Individual learning model

Individual learning is modeled as a reinforcement learning agent using $\epsilon$-greedy algorithm [32]. We denote estimated reward of an action $a$ at time $t$ as $Q(a, t)$. The reward estimation is updated based on the rewards $r(a, t)$ when $a$ is chosen using Eq (3).

$$Q(a, t + 1) = Q(a, t) + \beta[r(a, t) - Q(a, t)] \tag{3}$$

where $0 < \beta \leq 1$ is the step size which was set to 0.2 in our experiments. Eq (4) shows the $\epsilon$-greedy algorithm where the behavior with the highest estimated reward or a random behavior is chosen with the probabilities of $1 - \epsilon$ and $\epsilon$ respectively. We set $\epsilon = 0.1$ in our experiments.

$$a(t) = \begin{cases} \arg\max_a Q(a, t), & \text{based on the probability } 1 - \epsilon, \\ \text{random}(a), & \text{based on the probability } \epsilon. \end{cases} \tag{4}$$

### 4.3 Social learning models

Social learners copy the behavior of other individuals in the population based on a certain strategy [21]. We implement two social learning strategies as: success-based and conformist given in below:

$$a(t) = \begin{cases} \arg\max_a r(a, t - \tau), & \text{if success} - \text{based strategy}, \\ \arg\max_a h(a, t - \tau), & \text{if conformist strategy}. \end{cases}$$

where $r(a, t - \tau)$ and $h(a, t - \tau)$ denote the reward and frequency of an action $a$ at time $t - \tau$ with some latency $\tau$. Social learning is performed only when $t - \tau > 0$.

In success-based strategy, social learners copy the behavior of the individual with the best reward at $t - \tau$, and in conformist strategy, social learners copy the most frequent behavior in the population at time $t - \tau$.

## 4.4 Optimum distribution prediction uncertainty

Uncertainty of the environment ($U(t)$) is estimated based on the probability of sampling higher reward values from sub-optimum distributions. We refer to this probability as the optimum distribution prediction uncertainty (ODPU) and define as:

Let $\{X_{j_1}^1\}_{j_1}, \ldots, \{X_{j_k}^k\}_{j_k}$ be $k$ sets of normally distributed random variables, where the random variables of set $i$ are independently drawn from $\mathcal{N}(\mu_i, \sigma_i)$. All $k$ sets are finite, where $j_i$ represents an integer value such that $0 < j_i \leq N_i < \infty$ for all $i = 1, \ldots, k$. The notation $X_{(N_i)}^i$ is used to indicate the $N_i$-th order statistic. That is, the maximum of all random variables of a given set $\{X_{j_i}^i\}_{j_i}$.

Now, using the fact that the random variables are independently drawn from normal distributions, one can write the probability density function, $f$, and the cumulative distribution function, $F$, of the $N_{(i)}$-th order statistic as

$$f_{X_{(N_i)}^i}(x) = \frac{N_i}{\sigma_i}\phi\left(\frac{x - \mu_i}{\sigma_i}\right)\Phi\left(\frac{x - \mu_i}{\sigma_i}\right)^{N_i - 1}, \qquad F_{X_{(N_i)}^i}(x) = \Phi\left(\frac{x - \mu_i}{\sigma_i}\right)^{N_i}.$$

The optimum distribution prediction uncertainty is then be formulated by

$$\begin{aligned} \text{ODPU} \quad &= 1 - \mathbb{P}(X_{(N_1)}^1 \geq X_{(N_i)}^i; \quad \text{for all} \quad i \geq 2), \\ &= 1 - \int_{-\infty}^{\infty} f_{X_{(N_1)}^1}(y) \cdot \left(F_{X_{(N_2)}^2}(y) \cdot \cdots \cdot F_{X_{(N_k)}^k}(y)\right) \mathrm{d}y, \end{aligned}$$

which describes the probability of sampling higher values from the distributions with lower means relative to $\mu_1$.

## 4.5 Evolution of social learning

**4.5.1 Mathematical model.** This section provides our mathematical model for analysing the evolution of social learning strategies on the multi-armed bandit problem for number of arms $k = 2$. Let $a_i$, $i \in \{1, 2\}$ denote actions with corresponding payoffs $r(a_i, t)$ at time $t$.

The frequency of the individual learners in the population at time $t$ is denoted as $IL(t)$. We further distinguish two types of individual learners $A_1(t)$ and $A_2(t)$ to indicate the frequencies of the individuals that perform actions $a_1$ and $a_2$. The sum of the frequencies of the behaviors satisfy the following condition: $A_1(t) + A_2(t) = IL(t)$ for all $t \geq 0$.

We denote the frequencies of the social learners in the population as $SL(t)$ at time $t$. Overall, the sum of the individual and social learners in the population satisfy the following condition: $IL(t) + SL(t) = 1$ for all $t \geq 0$.

The change in the frequencies of $A_1$, $A_2$ and $SL$ are modeled using the replicator-mutator equation [44, 45] given as a system of coupled first order ordinary differential equations below

(dot notation represents time derivative, i.e. $\dot{x} = dx/dt$):

$$
\begin{cases}
\dot{A}_1(t) = \mathbf{F}(t)[\mathbf{I}^T(t) \circ \mathrm{col}_1(\mathbf{M})] - A_1(t)\psi(t), & t > 0, \\
\dot{A}_2(t) = \mathbf{F}(t)[\mathbf{I}^T(t) \circ \mathrm{col}_2(\mathbf{M})] - A_2(t)\psi(t), & t > 0, \\
\dot{SL}(t) = \mathbf{F}(t)[\mathbf{I}^T(t) \circ \mathrm{col}_3(\mathbf{M})] - SL(t)\psi(t), & t > 0, \\
A_1(t) = A_{1,0}, A_2(t) = A_{2,0}, SL(t) = SL_0, & t = 0.
\end{cases}
\tag{5}
$$

where $\mathbf{F}(t) := [f_{A_1}(t), f_{A_2}(t), f_{SL}(t)]$ is a row vector of fitness values, $\mathbf{I}(t) := [A_1(t), A_2(t), SL(t)]$ is a row vector of individual frequencies, $\circ$ denotes the element-wise multiplication operator, $\mathrm{col}_k()$ is a function that returns the $k$-th column of a matrix, and $\psi(t)$ is the average fitness of the population found as:

$$
\psi(t) := \mathbf{F}(t)\mathbf{I}^T(t)
\tag{6}
$$

The replication may not be perfect. The mutation probabilities are provided by $\mathbf{M}$ where $M_{ij}$ indicates the probability that type $j$ is produced by type $i$, and $\mathbf{M}_{i:}$ indicates row vector with an index of $i$. $\mathbf{M}$ is a row-stochastic matrix thus satisfies the following condition:

$$
\mathbf{M} \in \{\mathbf{A} \in \mathbb{R}_{\geq 0}^{3 \times 3} : \sum_{j=1}^{3} A_{ij} = 1, \ 1 \leq i \leq 3\}.
$$

In our experiments, we set mutation matrix $M$ as follows:

$$
M = \begin{bmatrix}
0.995 & 0 & 0.005 \\
0 & 0.995 & 0.005 \\
0.0025 & 0.0025 & 0.995
\end{bmatrix}
$$

which indicates mutation rate of 0.005 from the individual learners to social learner and vice versa.

Individual learners perform actions $a_1$ and $a_2$. However, they try the other action with a small frequency $\epsilon$ for exploration (e.g. analogous to $\epsilon$-greedy algorithm in reinforcement learning [32]). Thus they suffer from an exploratory cost. On the other hand, this may become useful for learning new action in case if the environment changes (i.e. change in the payoffs of the actions). Consequently, the fitness $f_{A_i}$ of type $A_i$ is found by the weighted average of payoffs obtained from performing different actions as shown in Eq (7).

$$
f_{A_i}(t) = (1 - \epsilon)r(A_i, t) + \epsilon r(A_j, t)
\tag{7}
$$

for all $i, j = \{1, 2\}$ where $i \neq j$.

Fitness of the the social learners $f_{SL}(t)$ updated based on a specific social learning strategy. We define four SLSs in following sections.

**Success-based:** The social learners copy the behavior of successful individual from a previous time $(t - \tau)$. Thus, the fitness values of the social learners equal to the reward received by performing the optimum action.

$$
f_{SL}(t) = r(a^*, t)
\tag{8}
$$

where $a^*$ denotes the optimum action $r(a^*, t)$ is its reward at time $t$.

**Conformist:** The social learners copy the behavior with the highest frequency in the population. We keep track of the behavior frequencies in the population and introduce some

latency represented as $\tau$. The frequencies of the behaviors performed by the social learners are also included into the model. We first show how the frequencies of the behaviors are computed and then define the fitness of conformist strategy.

Let $h(a_i, t)$ denote the frequencies of actions $a_i$ at time $t$. Furthermore, we define $H_{SL}(a_i, t)$ to denote the frequencies of the actions performed by the social learners. When $t = 0$, some portion ($H_{SL}(0)$) of the population consists of social learners, however they do not perform any action because the information of the frequencies of the actions in previous times ($t - \tau$) is not available. Therefore, we set the initial values when ($t - \tau <= 0$) as $H_{SL}(a_i, t) = 0$, $h(a_n, 0) = A_i(0)$, and $f_{SL}(0) = 0$.

When $t > \tau$, we update the frequencies of the actions as follows:

$$H_{SL}(a_i, t) = \begin{cases} 1, & \text{if } i = \arg \max_a h(a_i, t - \tau), \\ 0, & \text{otherwise.} \end{cases} \tag{9}$$

$$h(a_i, t) = A_i(t) + SL(t)h_{SL}(a_i, t) \tag{10}$$

Finally, the fitness of social learners, given in Eq (11), is found by the average payoffs of the social learners that perform each behavior type.

$$f_{SL}(t) = \sum_{i=1}^{n} h_{SL}(a_i, t)r(a_i, t). \tag{11}$$

**4.5.2 Evolutionary algorithm.** Algorithm 1 provides the pseudocode for the evolutionary algorithm [41] we use to analyse the evolution of a population of individual and social learners [28, 70]. Individuals are assigned one of these types randomly during the initialization process. In each generation, the individuals can perform their actions based on their type. Their fitness is computed based on their actions and used for the selection process for the next generation. We use only a mutation operator which alters the type of a selected individual with a small probability.

**Algorithm 1** the Evolutionary Algorithm for the evolution of social learning strategies.

```
1: procedure EA(ϵ, sls)        ▷ Evolution of individual and sls type
social learners
2:    // ϵ: exploration parameter of individual learning
3:    // sls: the type of social learning strategy (i.e. success-based
or conformist)
4:    t = 1      ▷ Generation counter t
5:    mr ≔ mutationRate
6:    I_t ≔ initializeIndividuals()       ▷Initial population
7:    while t ≤ T do
8:      for each i ∈ I_t do
9:        if (isIndividualLearner(i) or t = 1) then       ▷ Individual
Learning
10:           r_i(a_j, t) = individualLearning(ϵ)
11:           updateDecisionModel(r_i(a_j, t))        ▷ See Eq (3)
12:       else      ▷ Social Learning
13:           r_i(a_j, t) = socialLearning(sls)
14:           updateDecisionModel(r_i(a_j, t))
15:       end if
16:        f_i = updateFitness(r_i(a_j, t))
17:      end for
18:      I' ≔ select(I_t, F)
19:      I_{t+1} ≔ mutate(I', mr)
```

```
20:      t = t + 1
21:   end while
22: end procedure
```

The individual learners are modeled as reinforcement learning agents where they perform their behavior based on their model. Moreover, their models can be updated based on the rewards received as response to their behaviors. We use $\epsilon$-greedy algorithm, discussed in detail in Section 4.2 [32], to model the learning process of the individual learners.

The social learners on the other hand, perform their behaviors based on the behaviors of others. We model the same strategies, namely, conformist and success-based discussed in Section 4.5.1. In case of conformist strategy, the social learners select the behavior with maximum frequency in the population. For the success-based strategy, social learners copy the behavior of the individual with the best fitness value in the previous generation.

Depending on the outcome $r_i(a_j, t)$, that is the reward received by the $i$-th individual performing action $a_j$, their fitness values $f_i \in F$ are updated. We simply use the reward as the fitness of an individual $i$ at generation $t$ as: $f_i(t) = r_i(a_j, t)$. The average population reward $\psi(t)$ is the average of the fitness values of the individuals in the population: $\psi(t) = \frac{1}{m} \sum_i^m f_i(t)$ where $m$ is the number of individuals in the population.

Selection of the individuals for the next generation $t + 1$ is performed based on their fitness values at $t$. We use the *roulette wheel selection* (also known as *fitness proportionate selection* [41]) to simulate natural selection process. According to this scheme, $m$ number of individuals are selected based on the probability that is proportional to their fitness values as given below:

$$p_i(t) = \frac{f_i(t)}{\sum_{j=1}^{m} f_j(t)}, \tag{12}$$

where $p_i(t)$ is the probability of selecting individual $i$ from the population, and $f_i(t)$ is the fitness of the individual. The same individuals can be selected multiple times to construct the population for the next generation. There is no mating process involved. However, individuals are mutated by changing their type with a small probability controlled by the mutation rate ($mr$).

## 4.6 Meta-social learning

The meta-social learners can switch between individual and social learning during their lifetime. A generic algorithm for meta-social learning is provided in Algorithm 2. In this section, we provide the implementation details of all the algorithm variants used to control the meta-social learning strategies.

**Algorithm 2** Meta-social learning algorithm based on the environmental variables: environment change, uncertainty and conformity.

```
 1: procedure MSL(ε)      ▷ Run independently for each individual
 2:   // ε: exploration parameter of individual learning
 3:   // S: type of learning strategy ∈ {individual learning, success-
based or conformist strategy}
 4:   // t: discrete time counter
 5:   t = 1    ▷ Initial t
 6:   while t ≤ T do
 7:     [EC(t), C(t), U(t)] ≔ ContextEncoding(H, R)      ▷ Estimate the
context of the environmental
 8:     S ≔ MSL(EC(t), C(t), U(t))     ▷ Meta-social learner
 9:     if S = "individual learning" then     ▷ Individual Learning
10:       r_i(a_j, t) = individualLearning(ε)
11:       updateDecisionModel(r_i(a_j, t))     ▷ See Eq (3)
12:     else    ▷ Social Learning
13:       r_i(a_j, t) = socialLearning(S)
```

```
14:        updateDecisionModel(r_i(a_j, t))
15:    end if
16:    t = t + 1
17:  end while
18: end procedure
```

**4.6.1 Observation-based control.** The SL-EC-Conf-Unc algorithm uses three environmental variables, environment change ($EC(t)$), conformity ($C(t)$) and uncertainty ($U(t)$), to decide the type of learning strategy. Initially, and after an environment change, the algorithm uses individual learning strategy for exploration.

During individual learning, the conformity of the population and uncertainty of the environment are estimated and used for switching between conformity and success-based strategies as shown in Table 1.

Other variants in this class include SL-EC-Conf and SL-EC-Unc. These two variants use environment change but does not make use off full functionality of SL-EC-Conf-Unc. In case of SL-EC-Conf, individuals can use only individual learning and conformist strategies depending on the environment change and conformity, whereas, in case of SL-EC-Unc, they can use only individual and success-base strategies based on the environment change and uncertainty.

**4.6.2 Evolutionary control.** SL-GA (trained by the genetic algorithms) uses genetic algorithms (GAs [41]) to optimize the meta-social learning policies of the individuals for switching between the individual and social learning strategies. Shown in Table 2, we encode the type of strategy $s_j$ (i.e. individual learning, success-based or conformist) for a given state of the environmental variables, namely environment change, conformity and uncertainty. Since these variables can take binary values (see Section 4.6), there are 8 possible states (discrete variable) which can take three strategies. Thus, there are total of $3^8 = 6561$ possible distinct policies.

In addition to these discrete variables, we include two continuous variables into the genotype of individuals for determining the thresholds used in environment change ($th_{ec}$) and uncertainty ($th_u$). Consequently, the genotype of the individuals consist of 10 genes, 8 discrete and 2 continuous variables.

We use a standard GA with population size of 50 individuals, roulette wheel selection with four elites, 1-point crossover operator with 0.8 probability and a mutation operator which selects discrete genes with the probability of $1/(L - 2)$ where $L - 2$ is the length of the genotype excluding the genes that encode continuous variables, replaces one of three possibilities for the discrete genes, and performs Gaussian perturbation with zero mean and 0.1 standard deviation on the continuous genes.

**Table 1. The strategies implemented by the SL-EC-Conf-Unc based on the conformity and uncertainty.**

|  | Conformity ($C(t) = 1$) | Non-conformity ($C(t) = 0$) |
|---|---|---|
| **Low uncertainty ($U(t) = 0$)** | Conformist | Success-based |
| **High uncertainty ($U(t) = 1$)** | Conformist | Individual learning |

**Table 2. Possible states of the environment and their strategy assignments that can take one of the strategies as: Individual learning, success-based or conformist.**

| Environment Change | Conformity | Uncertainty | Strategy |
|---|---|---|---|
| 0 | 0 | 0 | $s_1$ |
| 0 | 0 | 1 | $s_2$ |
| . . . | . . . | . . . | . . . |
| 1 | 1 | 1 | $s_8$ |

We use a separate training environment for optimizing the meta-social learning policies using the GA (see Section S1.5 in S1 Text). The GA aims to maximize the fitness values of the policies which is computed by the median of the cumulative sum of the average population reward of 112 runs as follows:

$$f = \text{median}\left(\sum_{t=1}^{T} \psi_i(t)\right), \forall i = \{1, \ldots, 112\} \tag{13}$$

The GA process on the training environment is executed for 10 independent runs. We stop the evolutionary process if the algorithm fails to find a better fitness value for 20 subsequent generations. At the end of the runs, we select the best policy to be tested on the test environment reported in Results section.

S1 Diagram shows the best evolved policy over 10 independent GA runs (for the details of the optimization process see Section S1.6 in S1 Text). The evolutionary process is performed on a separate environment different than the environments we used for test in Results section. Note that the best evolved policy converged to the policy suggested by our analysis, and implemented by the SL-EC-Conf-Unc.

SL-NE (ANN based trained by neuroevolution) uses neuroevolution (NE) [43] approach to optimize artificial neural network (ANN) based policies. In NE, evolutionary algorithms are used for the optimization processes of the topologies and/or weights of the networks.

Illustrated S1 Fig, we use feed-forward ANNs with one hidden layer to perform individual learning, success-based and conformist social learning strategies. The input to the networks are the average and standard deviations of the estimated rewards of the actions, and frequencies of the individuals that perform each action (see Eq (2)). For two actions, we used 6, 12 and 3 neurons in the input, hidden and output layers respectively. We include an additional bias neuron (constant + 1) in input and hidden layers. Therefore, the total number of network parameters is 12(6 + 1) + 3(12 + 1) = 123.

We use genetic and differential evolution (DE) algorithms to optimize the weights of the networks by directly mapping them into the genotype of the individuals and representing them as real valued vectors. In both algorithms, we use a population of 50 individuals, and initialize the weights of the first generation randomly from the uniform distribution from range [−1, 1]. In case of the GA, we use roulette wheel selection with 5 elites, 1-point crossover operator with the probability of 0.8 and Gaussian mutation operator as: $\mathcal{N}(0, 0.1)$, that performs independent perturbation for each dimensions in the genotype.

In the case of the DE, we use "rand/1" mutation strategy and uniform crossover with parameters of $F = 0.5$ and $CR = 0.1$ respectively [42, 71].

Both algorithms aim to maximize the median of the total rewards, given in Eq (13), on the training environment provided in Section S1.5 in S1 Text. We run the GA and DE for 10 independent runs each, and use the ANN that achieved the best fitness value for the comparison in Results section.

**4.6.3 Multi-armed bandit control.**

1. SL-RL ($\epsilon$-greedy algorithm): the action-value based approach used in individual learning model (discussed in Section 4.2) is used for selecting the social learning strategy. In this case, the action space consists of performing one of the followings: individual learning, success-based and conformist strategies at time $t$. $Q(a, t)$ is the estimate reward of these learning approaches and updated based on the rewards received as shown in Eq (3). One of the actions are selected based on Eq (4).

2. SL-QL (Q-learning): we represent the estimate rewards of a certain action $a$ in a certain state of the environment $s$ as $Q(s_t, a_t)$ at time $t$. An action is selected based on the following:

$$a(t) = \begin{cases} \arg \max_a Q(s_t, a), & \text{based on the probability } 1 - \epsilon_{QL}, \\ \text{random}(a), & \text{based on the probability } \epsilon_{QL} \end{cases}$$

where $\epsilon_{QL}$ is exploration parameter for the Q-learning. Then, estimate rewards are updated based on the Bellman equation as follows:

$$Q(s_t, a_t) = Q(s_t, a_t) + \alpha(r(a, t) + \gamma \max_a Q(s_{t+1}, a) - Q(s_t, a_t))$$

where $\alpha$ and $\gamma$ are the learning rate and the discount factor.

Here, we use there environmental variables as states as: environment change $EC(t)$, conformity $C(t)$ and uncertainty $U(t)$. All of these variables are binary, thus, there are a total of eight states. There are three possible actions as: individual learning, success-based or conformist social learning strategies. We trained the SL-QL on the training environment provided in Section S1.5 in S1 Text. We performed experiments with different parameter settings of the algorithm and found that $\epsilon_{QL} = 0.2$, $\alpha = 0.01$ and $\gamma = 0$ value assignments provided the best results.

3. SL-UCB (upper confidence bound): to control the degree of the exploration, the equation for selecting actions is modified as follows:

$$a(t) = \arg \max_a \left[ Q(a, t) + c \sqrt{\frac{\ln t}{N(a, t)}} \right]$$

where $c$ is exploration parameter and $N(a, t)$ is the number of time $a$ is selected until time $t$. We use the same update rule for $Q(a, t)$ given in Eq (3).

In UCB selection, the square root term is the uncertainty in the estimate of $a$. While $N(a, t)$ increases, uncertainty terms decreases, whereas, while $N(a, t)$ keeps the same, uncertainty increases (since $t$ increases) [32].

### 4.6.4 Other baseline strategies

1. IL-Only (individual learning): individuals perform only individual learning throughout the processes.

2. SL-Rand (random strategies): individuals perform randomly one of individual learning, success-based and conformist social learning strategies with equal probability.

3. SL-Prop (proportional strategy selection): individuals perform success-based, conformist, individual learning strategies with probabilities of 0.45, 0.45 and 0.1 respectively.

4. SL-Conf (conformist with individual learning): individuals perform conformist and individual learning strategies with probabilities of 0.95 and 0.05 respectively.

5. SL-Succ (success-based with individual learning): individuals perform success-based and individual learning strategies with probabilities of 0.95 and 0.05 respectively.

## Supporting information

**S1 Text. Supporting information of the paper: Meta-control of social learning strategies.**
(PDF)

**S1 Diagram. Evolved Discrete part of the SL-GA policy that achieved the highest cumulative reward.** "Yes" and "No" indicate 1 and 0 states of the environmental variables shown in Table 2. Thresholds of the evolved rule for uncertainty and environment change is $th_u = 0.05$ and $th_{ec} = 0.15$.
(TIF)

**S1 Fig. Neuroevolution scheme used to optimize the social learning policies (SL-NE).** (a) Feed-forward artificial neural network topology with one hidden layer can take the average, standard deviations and frequencies of two actions $a_1$ and $a_2$ and decides to perform individual learning, success-based or conformist social learning strategies. (b) The weights of the networks between input and hidden layers ($W_{hi}$), and hidden and output layers ($W_{oi}$) are directly mapped to the genotype of the individuals and represented as real valued vectors. (c) Evolutionary algorithms are used to optimize the genotype of the individuals.
(TIF)

## Author Contributions

**Conceptualization:** Anil Yaman, Sang Wan Lee.

**Formal analysis:** Anil Yaman, Onur Çaylak, Sang Wan Lee.

**Funding acquisition:** Sang Wan Lee.

**Investigation:** Anil Yaman, Sang Wan Lee.

**Methodology:** Anil Yaman, Nicolas Bredeche, Onur Çaylak, Sang Wan Lee.

**Project administration:** Sang Wan Lee.

**Resources:** Sang Wan Lee.

**Software:** Anil Yaman.

**Supervision:** Sang Wan Lee.

**Validation:** Anil Yaman, Nicolas Bredeche, Joel Z. Leibo, Sang Wan Lee.

**Visualization:** Anil Yaman, Sang Wan Lee.

**Writing – original draft:** Anil Yaman, Sang Wan Lee.

**Writing – review & editing:** Anil Yaman, Nicolas Bredeche, Joel Z. Leibo, Sang Wan Lee.

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
