## [Decision Letter · Decision Letter 0]

2 Nov 2021

Dear Dr. Yaman,

Thank you very much for submitting your manuscript "Meta-control of social learning strategies" for consideration at PLOS Computational Biology.

As with all papers reviewed by the journal, your manuscript was reviewed by members of the editorial board and by several independent reviewers. In light of the reviews (below this email), we would like to invite the resubmission of a significantly-revised version that takes into account the reviewers' comments.

Please carefully address the criticism, especially of reviewer 1.

We cannot make any decision about publication until we have seen the revised manuscript and your response to the reviewers' comments. Your revised manuscript is also likely to be sent to reviewers for further evaluation.

Sincerely,

Arne Traulsen

Associate Editor

PLOS Computational Biology

Natalia Komarova

Deputy Editor

PLOS Computational Biology

Please carefully address the criticism, especially of reviewer 1.

Reviewer's Responses to Questions

**Comments to the Authors:**

Reviewer #1: Yaman and colleagues examined the performance of the social learning strategies in volatile and uncertain environments. Comparison between success-based and conformity-based social learning strategies based on the simulation data demonstrated that success-based social learning works better in the low-uncertainty environment. In contrast, the conformity-based strategy works better in high-uncertainty environments. Furthermore, they characterised an arbitration mechanism (i.e., meta-control) of the learning strategies that resolves environmental uncertainty with minimal exploration cost. I believe this study addressed the critical issue and would potentially provide a unified framework of social learning that spans evolutionary biology, psychology, neuroscience, and machine learning.

One of my major concerns is the conceptual validity of success-based social learning. The success-based strategy requires an agent to keep track of the others’ actions as well as their reward outcomes. If each agent has the cognitive capacity of doing so, s/he can use another form of social learning: that is, learning from others' outcomes (i.e., observational learning). Indeed, recent studies in social neuroscience have suggested that humans combine learning from others’ actions and that from others' rewards. It would be great if the authors could test for the case.

I am also concerned with the robustness of their conclusions. The simulations apparently have a lot of parameters to be determined in an arbitrary way (e.g., |mu_1 – mu_2|, sigma, th_ec, tc_u). In other words, the overall conclusions of the study may be changed with different sets of parameter values. For instance, the mutation rate is known to have the potential to dramatically change the evolutionary phenomena from the stational to oscillatory and chaotic dynamics (Nowak & Sigmund, PNAS 1993). The authors could perform exhaustive sensitivity analyses in order to check the robustness of their findings with respect to the choice of parameter values. Furthermore, I am wondering what happens in the case of binary reward outcome (reward or no-reward), as previous studies have mainly focused on binary reward.

To explore the evolutionary dynamics, researchers often examine the evolutionary stability (i.e., whether the population dominated by the strategy can resist the invasion of a small number of individuals who employ another strategy), as well as the basin of attraction (i.e., how much the strategy dominates the population depending on the different initial distributions of the strategies). The authors could address those points.

In economics, social learning has been studied in the context of 'information cascade', which demonstrates an adverse effect of social learning. Furthermore, the authors pointed out the possibility that exploration is costly. One fascinating question is who incurs the cost and who free-rides others’ exploration (e.g., see Bolton and Harris, Econometrica 1999). I am wondering if the present study has implications for those issues.

The key concept of the paper is the optimum distribution prediction uncertainty (ODPU). I believe it would be helpful to explain the concept in a concise way in the main text (e.g., in the Introduction or Section 2.2). I was a little bit confused with the concepts of uncertainty, risk (sigma) and volatility. For example, for me, it was difficult to understand the statement like "we hypothesised that the high uncertainty in the environment would make it hard to identify successful individuals (lines 138-139)" without the clear definition of uncertainty.

The legends of Figures 2, 4 and 5 could be more informative. I would like the authors to provide more detailed information in the legends, so that the naïve readers understand, for example, how to read out Fig 2a and the critical difference (CD) diagrams and so on.

In some parts, the authors reported p-values of the statistical test. Does it make sense to perform the statistical test on the simulated data (where the sample size is not meaningful)?

Reviewer #2: In this work, the authors propose an approach to social learning, which they name meta-social learning.

They justify their approach on studies on brain skills on learning strategies and cognitive-behavioral science. The authors deal with a trade-off between environmental uncertainty and performance-cost rate by a model that explores individual learning and two different social learning strategies, namely success-based and conformist. These strategies are well-known in the field of social learning.

To do this, the authors use a common set-up, the multi-armed bandit, implementing two different dynamics: replicators and agent-based.

Via numerical simulations, they find that in uncertain environments, the conformist strategy performs better than the success-based strategy, In general, none of the three strategies achieve an optimal policy for lifetime learning. Therefore, the authors propose a mixed strategy, the so-called meta-social learning strategy, to fit in an environment characterized by both volatility and uncertainty. The proposed meta-social learning strategy uses estimated uncertainty to arbitrate between the three pure strategies (individual learning, success-based and conformist).

They successfully tested their model with a large set of different well-known algorithms.

I like the paper as it is and recommend its publication.

A couple of minor remarks (optional):

I find confusing the Panel a of Figure 1. I am not sure how to read it (at least, I find much easier the main text to understand the model).

The authors may consider taking a look to this paper:

Cardoso, F.M. et al. Dynamics of heuristics selection for cooperative behaviour. New Journal of Physics, 22(12), p.123037. (2020)

Reviewer #3: This paper is about discussing whether to learn individually or to imitate other strategies in a group, in the uncertainty of the environment.

First of all, at the beginning, I wanted it to say that individual learning here means e-greedy RL, see 4.2, should be clearly stated. Also what we are really doing in social learning is found in 4.3, should be mentioned. I couldn't figure that out and had a hard time reading the analogy the whole time.

On top of that, the results are not very surprising. What happens when the degree of uncertainty is varied continuously? The definition of uncertainty is given too easy or to formalistic. Agents should resolve the uncertainty or to infer the behind. The present approach is too much top down (or GOFAI) and I didn't learn anything from here.

**Have the authors made all data and (if applicable) computational code underlying the findings in their manuscript fully available?**

Reviewer #1: Yes

Reviewer #2: Yes

Reviewer #3: Yes

PLOS authors have the option to publish the peer review history of their article (what does this mean?). If published, this will include your full peer review and any attached files.

Reviewer #1: No

Reviewer #2: No

Reviewer #3: No
---

## [Decision Letter · Decision Letter 1]

28 Jan 2022

Dear Dr. Yaman,

We are pleased to inform you that your manuscript 'Meta-control of social learning strategies' has been provisionally accepted for publication in PLOS Computational Biology.

This was a difficult case, as one reviewer recommended rejection and another one acceptance, but both reviews were somewhat limited in scope and hard to address. Thus, for the revision I only invited the reviewer who had originally recommended a major revision, as I thought their report was very balanced and constructive. That reviewer is now recommending acceptance. 

Best regards,

Arne Traulsen

Associate Editor

PLOS Computational Biology

Natalia Komarova

Deputy Editor

PLOS Computational Biology

Reviewer's Responses to Questions

**Comments to the Authors:**

Reviewer #1: The authors have adequately addressed all the concerns.

**Have the authors made all data and (if applicable) computational code underlying the findings in their manuscript fully available?**

Reviewer #1: Yes

PLOS authors have the option to publish the peer review history of their article (what does this mean?). If published, this will include your full peer review and any attached files.

Reviewer #1: No

---

## [Editor Report · Acceptance letter]

17 Feb 2022

PCOMPBIOL-D-21-01641R1 

Meta-control of social learning strategies

Dear Dr Yaman,

I am pleased to inform you that your manuscript has been formally accepted for publication in PLOS Computational Biology. Your manuscript is now with our production department and you will be notified of the publication date in due course.

With kind regards,

Olena Szabo
